# Heterogenous Genetic, Clinical, and Imaging Features in Patients with Neuronal Intranuclear Inclusion Disease Carrying *NOTCH2NLC* Repeat Expansion

**DOI:** 10.3390/brainsci13060955

**Published:** 2023-06-15

**Authors:** Yusran Ady Fitrah, Yo Higuchi, Norikazu Hara, Takayoshi Tokutake, Masato Kanazawa, Kazuhiro Sanpei, Tomone Taneda, Akihiko Nakajima, Shin Koide, Shintaro Tsuboguchi, Midori Watanabe, Junki Fukumoto, Shoichiro Ando, Tomoe Sato, Yohei Iwafuchi, Aki Sato, Hideki Hayashi, Takanobu Ishiguro, Hayato Takeda, Toshiaki Takahashi, Nobuyoshi Fukuhara, Kensaku Kasuga, Akinori Miyashita, Osamu Onodera, Takeshi Ikeuchi

**Affiliations:** 1Department of Molecular Genetics, Brain Research Institute, Niigata University, Niigata 951-8585, Japan; yusranadyfitrah7@gmail.com (Y.A.F.); neurohiguchi@gmail.com (Y.H.); nhara@bri.niigata-u.ac.jp (N.H.); ken39@bri.niigata-u.ac.jp (K.K.); miyashi2020@bri.niigata-u.ac.jp (A.M.); 2Department of Neurology, Brain Research Institute, Niigata University, Niigata 951-8585, Japan; tokutaketaka@yahoo.co.jp (T.T.); masa2@bri.niigata-u.ac.jp (M.K.); sabatama@me.com (T.T.); ymanruna@gmail.com (A.N.); shin.m0203@gmail.com (S.K.); tsuboguchi@bri.niigata-u.ac.jp (S.T.); nagareume.cla01@gmail.com (M.W.); junki142235@gmail.com (J.F.); shoando3@gmail.com (S.A.); hayashihideki827@gmail.com (H.H.); tishiguro211@gmail.co (T.I.); onodera@bri.niigata-u.ac.jp (O.O.); 3Department of Neurology, Joetsu General Hospital, Joetsu 943-0172, Japan; fukuhara@joetsu.jp; 4Department of Neurology, Sado General Hospital, Sado 952-1209, Japan; aab91360@pop13.odn.ne.jp; 5Department of Neurology, Tsubame Rosai Hospital, Tsubame 959-1228, Japan; tomoesatochapon@gmail.com; 6Department of Neurology, Niigata City General Hospital, Niigata 950-1197, Japan; yiwafuchi2@gmail.com (Y.I.); satoaki@hosp.niigata.niigata.jp (A.S.); 7Department of Neurology, Tsukuba University, Tsukuba 950-1197, Japan; hayato01181984@gmail.com; 8Department of Neurology, Kido Hospital, Niigata 950-0862, Japan; toshiaki4649@gmail.com

**Keywords:** neuronal intranuclear inclusion disease, GGC repeat expansion, *NOTCH2NLC*, clinical presentations, neuroimaging, genotype–phenotype correlation

## Abstract

Neuronal intranuclear inclusion disease (NIID) is a neurodegenerative disorder that is caused by the abnormal expansion of non-coding trinucleotide GGC repeats in *NOTCH2NLC*. NIID is clinically characterized by a broad spectrum of clinical presentations. To date, the relationship between expanded repeat lengths and clinical phenotype in patients with NIID remains unclear. Thus, we aimed to clarify the genetic and clinical spectrum and their association in patients with NIID. For this purpose, we genetically analyzed Japanese patients with adult-onset NIID with characteristic clinical and neuroimaging findings. Trinucleotide repeat expansions of *NOTCH2NLC* were examined by repeat-primed and amplicon-length PCR. In addition, long-read sequencing was performed to determine repeat size and sequence. The expanded GGC repeats ranging from 94 to 361 in *NOTCH2NLC* were found in all 15 patients. Two patients carried biallelic repeat expansions. There were marked heterogenous clinical and imaging features in NIID patients. Patients presenting with cerebellar ataxia or urinary dysfunction had a significantly larger GGC repeat size than those without. This significant association disappeared when these parameters were compared with the total trinucleotide repeat number. ARWMC score was significantly higher in patients who had a non-glycine-type trinucleotide interruption within expanded poly-glycine motifs than in those with a pure poly-glycine expansion. These results suggested that the repeat length and sequence in *NOTCH2NLC* may partly modify some clinical and imaging features of NIID.

## 1. Introduction

Neuronal intranuclear inclusion disease (NIID) is a progressive neurodegenerative disease, which is clinically characterized by a broad spectrum of clinical presentations [1]. The formation of eosinophilic intranuclear inclusions in the central and peripheral nervous system as well as other organs is a pathological hallmark of NIID [2,3]. Depending on familial occurrence, NIID is classified into the sporadic and autosomal dominant forms [1]. Sporadic occurrence was reported to be approximately 67% in NIID [1]. The average age at onset in patients with NIID was reported to be 56 years (range: 16–76 years) [1]. Among its variable clinical presentations are cognitive decline, encephalitic episodes, parkinsonism, cerebellar ataxia, autonomic dysfunction, and muscle weakness [1,2]. Clinical diagnosis of NIID has been difficult because of substantial heterogenous clinical presentations. Brain MRI typically shows extensive white matter lesions on fluid-attenuated inversion recovery (FLAIR) and T_2_-weighed (T_2_-WI) images and a high-intensity lesion along the U-fiber on diffusion-weighed images (DWI) [1]. These MRI signs often alert clinicians to a possible diagnosis of NIID. Despite this knowledge, the pathogenesis underlying NIID has not yet been resolved in detail; thus, NIID remains incurable.

A GGC repeat expansion in the 5’ untranslated region (UTR) of *NOTCH2NLC* has been identified as a causative mutation in patients with NIID [4,5,6]. Although the exact prevalence of genetically confirmed NIID has not been documented, over 400 patients have been reported in the literature. Patients with NIID carrying the GGC expansion of *NOTCH2NLC* have been predominantly reported among Asian populations [4,5,6] but rarely in European ones [7]. NIID due to the GGC repeat expansion of *NOTCH2NLC* accounted for 13% and 20% of patients with genetically undetermined adult-onset leukoencephalopathies in Japanese and Taiwanese cohorts, respectively [8,9]. NIID is one of major causes underlying adult-onset leukoencephalopathies in Asians. Expanded GGC repeats are relatively stable between generations [10]. Based on large series of genetically confirmed NIID patients, Tian et al. reported that NIID can be divided into four subgroups: dementia-dominant, movement-disorder-dominant, muscle-weakness-dominant, and paroxysmal-symptom-dominant [10]. As the GGC repeat length was significantly higher in the muscle-weakness-dominant subgroup than in the other subgroups, the GGC repeat size may differently affect the various clinical subtypes of NIID [8].

Despite the increased number of reported patients with genetically confirmed NIID, the relationship between repeat length and sequence in *NOTCH2NLC* and the diverse phenotypes of NIID remains unclear. Thus, we aimed to demonstrate a broad spectrum of genetic, clinical, and imaging features as well as their possible associations in 15 Japanese patients with NIID carrying a *NOTCH2NLC* repeat expansion. By this analysis, we report heterogenous genetic, clinical, and imaging findings in patients with NIID. Furthermore, our results suggested that repeat length and sequence in *NOTCH2NLC* may modify the phenotype of NIID.

## 2. Materials and Methods

### 2.1. Patients

Fifteen patients who met the inclusion and exclusion criteria were recruited from 6 medical institutes between April 2018 and November 2022. All the patients underwent a neurological examination, routine blood testing, and neuroimaging evaluation. In a recruitment process, the patients were recruited regardless of gender. As a result, we recruited 10 female and 5 male subjects. Inclusion criteria included (1) age at onset ≥ 20 years, (2) presence of the white matter lesions shown in MRI or CT scans, and (3) the presence of neurological, cognitive, or psychiatric symptoms. To increase the specificity, subjects with the following conditions were excluded: (1) white matter lesions secondary to demyelinating disease, infection, toxins, or neoplasm; (2) past history of cerebral ischemic infarctions or intracranial hemorrhage; and (3) the presence of known mutations of white matter diseases such as *CSF1R*, *NOTCH3*, or *HTRA1*. Skin biopsy was performed in 12 out of 15 patients prior to genetic testing. Details of clinical presentations in Pt 6 were previously reported [11] and those of Pt 15 will be published elsewhere. This study was approved by the Ethics Committee of Niigata University (G2015-0849, 2019-0239). Written informed consent was obtained from the patients or their caregivers.

### 2.2. Repeat-Primed and Amplicon-Length PCR Analyses

Genomic DNA was extracted from peripheral blood using an automated DNA extraction system (QuickGene-Auto240L, Kurabo, Japan). Repeat expansion in *NOTCH2NLC* was examined by repeat-primed PCR, followed by electrophoresis on a 3500xl Genetic analyzer (ThermoFisher Scientific, Waltham, MA, USA), as described previously [6]. Briefly, PCR was performed in a total volume of 10 μL of reaction solution containing 0.25U PrimeSTAR GXL DNA Polymerase; 1 × PrimeSTAR GXL Buffer; 200 µM of dATP, dTTP, dCTP (Takara Bio, Shiga, Japan), and 7-Deaza-2′-deoxy-guanosine-5′- triphosphate (Sigma-Aldrich, St. Louis, MO, USA); 5% dimethyl sulfoxide (Sigma-Aldrich); 1 M betaine (Sigma-Aldrich); 0.3 µM of each primer mix; and 100 ng of genomic DNA. The presence of a sawtooth pattern in the electropherogram was regarded as an abnormal repeat expansion. *NOTCH2NLC* repeat size was further determined by fluorescence amplicon-length PCR as described previously [6]. GeneMapper software (ThermoFisher Scientific) was used to determine GGC repeat length.

### 2.3. Long-Read Sequencing

Sequencing libraries were prepared from 1.5 μg of genomic DNA using the Ligation Sequencing Kit SQK-LSK109 (Oxford Nanopore Technologies, Oxford, UK) according to the manufacturer’s instructions. Each library was sequenced on a PromethION R9.4.1 flow cell (Oxford Nanopore Technologies) with pore washing and library reloading to increase the sequencing yield. Base-calling was performed in real time using the MinKNOW software installed on a PromethION 24 device (Oxford Nanopore Technologies). This genetic analysis yielded an average of 96.1 gigabases of sequence data from each sample.

To determine the number of GGC tandem repeats in *NOTCH2NLC*, we used the *CharONT* v1.1.0 pipeline [12]. First, we ran the Extract_Xdrop_alignments.sh script to map the sequenced reads to the human reference genome hg38 using minimap2 [13]. We collected the reads mapped around the *NOTCH2NLC* repetitive region defined in a bed file (chr1:149,390,402–149,391,242). Next, we ran the CharONT.R script to make consensus sequences for both alleles. Finally, the polished consensus sequences were analyzed by Tandem Repeats Finder to identify repeat length [14].

### 2.4. Clinical Assessments

Clinical histories of the patients, including demographic information, family history, age at onset and examination, disease duration, and clinical symptoms, were obtained using standardized case report forms. We examined the presence and absence of muscle weakness, parkinsonism, tremors, episodic encephalopathy, ataxia, sensory disturbances, headaches, visual disturbances, psychiatric symptoms, bladder dysfunction, and cognitive impairments. Neurological examinations were performed by board-certified neurologists. Family history was defined on the presence of clinical symptoms related to NIID in first or second relatives. The Mini-Mental State Examination (MMSE), which is a screening cognitive battery with maximum score of 30, was used to evaluate general cognitive function.

### 2.5. Laboratory Tests

The cerebrospinal fluid (CSF) examination and nerve conduction velocity test were performed in a subset of the patients. CSF was obtained by standard lumber puncture. The initial tube for CSF collection was examined for routine cellular and biochemistry examination. The subsequent CSF samples were collected into polypropylene tubes, followed by freezing, and then shipped to Niigata University. CSF was aliquoted at a volume of 0.5 mL and stored at −80 °C until the measurement. All CSF analyses were conducted in duplicate by experienced laboratory personnel blinded to the clinical diagnosis. CSF biomarkers including amyloid-β 40 (Aβ40), Aβ42, total tau (t-tau), phosphorylated tau at threonine 181 (p-tau181), and neurofilament light chain (NfL) were examined as previously described [15]. AT(N) classification was performed as previously described according to the abnormal CSF values of Aβ42 (>359.6 pg/mL), Aβ42/40 ratio (>0.072), p-tau (>30.6 pg/mL), ant t-tau (>105.3 pg/mL), and NfL (>2650 pg/mL) [15,16].

### 2.6. MRI Evaluation

All participants underwent MRI scans using a 1.5 or 3 tesla scanner at clinical institutes visited by the patients. The T_1_-weighted, T_2_-weighted, fluid-attenuated inversion recovery (FLAIR), and diffusion-weighted image (DWI) were performed. DICOM images were sent anonymously to Niigata University, where two board-certified neurologists independently reviewed the imaging findings. An age-related white matter change (ARWMC) score was determined according to a previous study [17].

### 2.7. Statistical Analysis

For descriptive statistics, the mean and standard deviation (SD) were used for continuous variables. The Mann–Whitney U test for comparing two groups were applied if there was any non-normally distributed data. Correlation analysis between two data sets was performed using Spearman’s rank test. The statistical significance level was set at *p* < 0.05. The GraphPad PRISM 9 software was used for all statistical analyses (GraphPad Software Inc., San Diego, CA, USA).

## 3. Results

### 3.1. Genetic Findings

Repeat-primed PCR analysis revealed a sawtooth pattern in 15 patients clinically suspected as having NIID, suggesting the presence of expanded repeats in *NOTCH2NLC*. Representative positive and negative results of repeat-primed PCR are shown in Appendix A. To determine the repeat size, we performed amplicon-length PCR (Appendix A). The repeat length of the normal allele was 8.5 ± 1.5 (mean ± SD; range: 7–10) and that of the expanded repeat size was 92 ± 13 (range: 71–110). Expanded repeats were undetectable in four patients by amplicon-length PCR even though repeat-primed PCR revealed a sawtooth pattern. In Pt 7, the analysis of the amplicon-length PCR showed a biallelic repeat expansion (Appendix A).

To determine the repeat length and sequence, we performed long-read sequence using a Nanopore sequencer. The expanded repeat size of any trinucleotide sequence was 138 ± 69 (range: 94–361) (Table 1). The expanded GGC repeat size was 114 ± 36 (range: 44–193). The poly-glycine repeat encoded by GGC, GGA, or GGG was 116 ± 36 (range: 47–197). Pt 2 showed a pure GGC repeat expansion and other patients carried other trinucleotide sequences including GGA, AGC, and GAC in addition to expanded GGC repeats (Table 1, Appendix A). The most frequent trinucleotide insertion was GGA at the 3′ end of GGC repeats (12/15, 80%), as observed in the normal allele (Appendix A). Expanded poly-glycine repeats were interrupted by the non-glycine-type trinucleotide sequence in 47% of patients (7/15; Table 1, Figure 1, Appendix A). We identified biallelic repeat expansions in Pt 7 (109 and 117 repeats) and Pt 14 (135 and 361 repeats) (Figure 1). The expanded allele with 361 repeat units in Pt 14 showed the unique trinucleotide sequence: (TTA)_76_(CTA)_22_(TCA)_12_(TTT)_8_(ATA)_8_(AAT)_6_(GAT)_125_ (GAC)_20_(GAG)_1_(ATC)_1_(ACC)_4_(CCA)_15_(CAC)_3_(ACA)_2_(TAC)_2_(TCA)_2_(GGA)_1_(GGG)_2_(GGC)_44_(GCT)_1_(CCT)_1_(CAG)_1_(ACT)_1_(TGC)_1_CTG)_1_(ATG)_2_ (Figure 1).

### 3.2. Clinical Findings

The clinical characteristics of the 15 patients are summarized in Table 2. The age at onset was 64.1 ± 8.1 years (range: 50–74 years). The disease duration was 8.1 ± 6.7 years (range: 2–28 years). Two (13.3%) of 15 patients with NIID had a family history. We classified the patients into four subtypes according to a previous report [10]. The most frequent subtypes were dementia-dominant (6/15, 40%) and paroxysmal-symptom-dominant (6/15, 40%) followed by movement-disorder-dominant (2/15, 13%). The clinical phenotype of Pt 12, who exhibited an autonomic-dysfunction-dominant presentation, could not be classified into any particular subtype. The skin biopsy results were positive for p-62-positive intranuclear inclusions in all patients (n = 12).

The most frequent initial symptom was encephalitic episodes (4/15, 27%), followed by movement disorders (tremor and gait disturbance; 3/15, 20%) and dementia (3/15, 20%). Encephalitic episodes included stroke-like episodes, epileptic seizures, and consciousness disturbances. During the disease course, diverse clinical presentations were observed, with cognitive decline being the most frequent symptom (11/15, 73%). The MMSE score was 20.4 ± 7.2 (range: 7–28). Other frequent clinical features were hyporeflexia (10/14, 71%), consciousness disturbance (9/15, 60%), cerebellar ataxia (9/15, 60%), dysarthria (8/15, 53%), gait disturbance (8/15, 53%), and urinary disturbances (7/15, 47%) such as incontinence and urinary retention.

The clinical presentation of Pt 7 and 14 who had biallelic repeat expansions were comparable to those of other NIID patients heterozygous for repeat expansions. The age at onset was 50 years in Pt 7 and 60 years in Pt 14, and the clinical subtype in both biallelic patients was the paroxysmal-symptom-dominant-type. Pt 7 had a family history and a slow progression with a disease duration of 28 years.

CSF biomarkers were analyzed in three patients with a dementia-dominant subtype (Table 3). Although Aβ42 and Aβ42/40 levels were within the normal range, those of p-tau, t-tau, and NfL were increased. Based on the AT(N) classification, three patients with NIID were assigned to the A–T+(N)+ profile [15].

### 3.3. Neuroimaging Findings

MRI findings are summarized in Table 4. Cerebral white matter lesions were observed in all patients (Figure 2). In DWI, high-intensity lesions were frequently observed along the U-fiber (14/15, 93%), even extending to the posterior lobe in some patients (6/15, 44%) (Figure 2). In FLAIR images, high-intensity lesions in the corpus callosum were observed in 93% (14/15) of patients. In FLAIR images, high-intensity lesions in corpus callosum were observed in 93% (14/15) of patients. Some patients showed cerebellar abnormalities on MRI, including cerebellar atrophy (11/15, 73%), middle cerebellar peduncle lesions (3/15, 33%), and paravermal lesions (5/15, 33%) (Figure 2). Then, we semi-quantified the white matter changes by ARWMC scores, obtaining a score of 17.1 ± 4.4 (range: 8–24; Table 4).

### 3.4. Genotype–Phenotype Correlations

Next, we investigated correlations between the size of *NOTCH2NLC* repeat expansions and clinical/imaging features in patients with NIID. Repeat length was evaluated by the total repeat size of any trinucleotide sequence, cumulative GGC, or poly-glycine repeat units. There were no significant correlations between repeat length and age at onset, MMSE score, or ARWMC score (Appendix A). Patients presenting with clinical symptoms of cerebellar ataxia or urinary dysfunction had a significantly larger GGC repeat size than those without (Table 5). However, these significant differences disappeared when these parameters were compared with the total repeat size of any trinucleotide (Appendix A). In addition, a significantly higher ARWMC score was observed in patients who had a non-glycine-type trinucleotide interruption such as AGC, GAC, CGC, GAA, CAC, CCA, CCC, CCG, ACG, and GCA within expanded poly-glycine motifs compared to those with a pure poly-glycine expansion encoded by GGC, GGA, and GGG (Figure 3).

## 4. Discussion

Using repeat-primed and amplicon-length PCR followed by Nanopore long-read sequencing, we identified a GGC repeat expansion of *NOTCH2NLC* in 15 Japanese patients clinically suspected as having NIID. The repeat sequence and size varied among patients with NIID. The expanded repeat size was 94–361. One patient had pure expanded GGC repeats. In other patients, other trinucleotide repeats such as GGA, AGC, and GAC were inserted in the expanded GGC repeat motifs as previously reported [4,5,6]. We characterized genetic, clinical, and imaging features in patients with NIID and reported several noteworthy findings in this study.

We identified two NIID patients (Pt 7 and Pt 14) as compound heterozygotes for the repeat expansions. One expanded allele in Pt 14 contained a total of 361 repeats containing an expanded (TTA)_76_ (GAT)_125_ (GGC)_44_ motif, which was not previously reported. The age at onset of patients carrying biallelic expansions was 50 years for Pt 7 and 60 years for Pt 14. Both exhibited a paroxysmal-symptom-dominant phenotype characterized by consciousness disturbances, encephalitis episodes, and cognitive decline. A previous study reported that patients with biallelic GGC repeat expansions exhibited a typical dementia-dominant NIID phenotype [18]. Kameyama et al. argued that biallelic repeat expansions likely show a dominant effect on the phenotype of NIID. Our findings on compound heterozygous patients support their notion because the phenotypes of our patients with biallelic expansions were comparable to those of patients heterozygous for the expansion.

On the basis of large series of genetically confirmed NIID, Tian et al. reported that NIID can be classified into four subtypes. In their study, the most frequent subtype was dementia-dominant (38%), followed by movement-disorder-dominant (26%), paroxysmal-symptom-dominant (25%), and muscle-weakness-dominant (12%) [10]. The clinical presentations of each subtype may overlap [10]. In this study, the dementia-dominant and paroxysmal-symptom-dominant subtypes were the most frequent. Pt 12 who exhibited an autonomic-dysfunction-dominant subtype with urinary dysfunction could not be readily classified into any subtype. Since autonomic dysfunction is commonly reported [10], an autonomic-dysfunction-dominant subtype may be another subtype of NIID.

We determined CSF biomarkers in three patients with NIID (Table 3). All three had an A–T+(N+) profile according to the research framework [16]. We found markedly elevated NfL levels in CSF, suggesting the presence of neurodegeneration in NIID. Consistent with our result, Chen et al. reported elevated plasma NfL levels in patients with NIID [19]. Unexpectedly, the p-tau levels are elevated in the CSF of patients with NIID. The p-tau levels in CSF are elevated in patients with Alzheimer’s disease (AD) spectrum showing Aβ accumulation in the brain, which accelerates p-tau secretion from damaged neurons [15,20]. However, the Aβ42 level and the Aβ42/40 ratio were not changed in our NIID patients. Since Aβ and phosphorylated tau depositions in the brain have not been linked to NIID, the mechanism underlying the increased p-tau level in NIID may differ from that of AD. Consistent with our results, Kurihara et al. recently demonstrated that 75% (9/12) of patients with NIID showed A-T+ profiles with increased p-tau and normal Aβ42 levels in CSF [21]. They speculated that the p-tau increase in CSF of patients with NIID may be caused by an enhanced secretion of p-tau from neurons through an unknown mechanism, rather than a nonspecific increase due to acute neurodegeneration.

High-intensity lesions along the U-fiber on DWI and marked cerebral white matter lesions on FLAIR have been reported as characteristic MRI features in patients with NIID [1]. The white matter lesions are typically confluent and bilateral with a predominance of the frontal lobe. These MRI features were very frequently (>90%) observed in our patients with NIID (Table 4). In addition, high-intensity lesions were frequently observed in the corpus callosum in patients with NIID. NIID should be regarded as one of the important causes for patients with adult-onset leukoencephalopathies. Imaging abnormalities of the cerebellum are also frequently detected by MRI in such patients [22]. Sugiyama et al. reported that MRI showed cerebellar atrophy (8/8 patients), high-intensity lesions in the middle cerebellar peduncle (4/8), and paravermal abnormal signals (6/8) in patients with NIID [22]. In line with those results, cerebellar atrophy was observed in 73% (11/15), and middle cerebellar peduncle and paravermal lesions were observed in 33% (3/15) and 44% (6/15) of patients, respectively, in this study.

In addition, we showed that patients presenting with ataxia or urinary dysfunction had a significantly larger GGC repeat length than those without (Table 5). These correlations disappeared if these parameters were compared with the total repeat size of any trinucleotide (Appendix A). This finding suggests that GGC or poly-glycine repeat size may have a stronger impact on some clinical features in NIID than the total trinucleotide repeat number. This may be supported by a previous report in which the translation of GGC repeat expansions into a toxic poly-glycine stretch plays a pathological role in NIID [23]. Moreover, previous research has reported that GGA disruptions may be associated with a muscular-weakness-dominant subtype [5] and a younger age at onset [19]. We showed that the presence of a non-glycine-coding-type trinucleotide sequence interruption may modify the severity of the white matter changes as determined by ARWMC. These results suggest that repeat length and sequence in *NOTCH2NLC* may partly modify the phenotype of NIID.

Our study has several limitations. First, we performed a cross-sectional study involving a relatively small number of patients with NIID. This may make this study underpowered to perform genotype–phenotype correlations. More studies are needed to validate our findings. It would be important to collect longitudinal data of patients with NIID to understand the natural history of NIID. Second, the retrospective design of the study might not allow us to fully illustrate the clinical characteristics of NIID. Future prospective studies are required to fully understand the broad phenotypic spectrum of NIID. Third, there may be recruitment bias in this study, as we recruited patients with white matter lesions on MRI. Patients with muscle weakness or peripheral neuropathy not showing MRI abnormalities may be underestimated. Last, although we determined the length and sequence of repeat expansions using a Nanopore long-read sequencer, we did not examine the methylation status of *NOTCH2NLC*. Recent studies have suggested that the methylation status affects the expression levels of *NOTCH2NLC* if the repeat sequence is highly expanded [17,24,25]. Thus, analysis of methylation status in our patients warrants further investigation.

## 5. Conclusions

We identified 15 NIID patients with *NOTCH2NLC* GGC repeat expansions in whom genetic, clinical, and imaging features were markedly variable. Our study suggests that repeat length and sequence in *NOTCH2NLC* may partly modify the phenotype of NIID. The present findings demonstrated the potential of repeat-primed and amplicon-length PCR followed by a long-read sequence for the accurate genetic diagnosis of NIID. Ensuring a correct diagnosis by genetic analyses is important for the better management of patients with NIID. A further study with a larger sample size of NIID patients is required to validate our findings.

## Figures and Tables

**Figure 1 brainsci-13-00955-f001:**
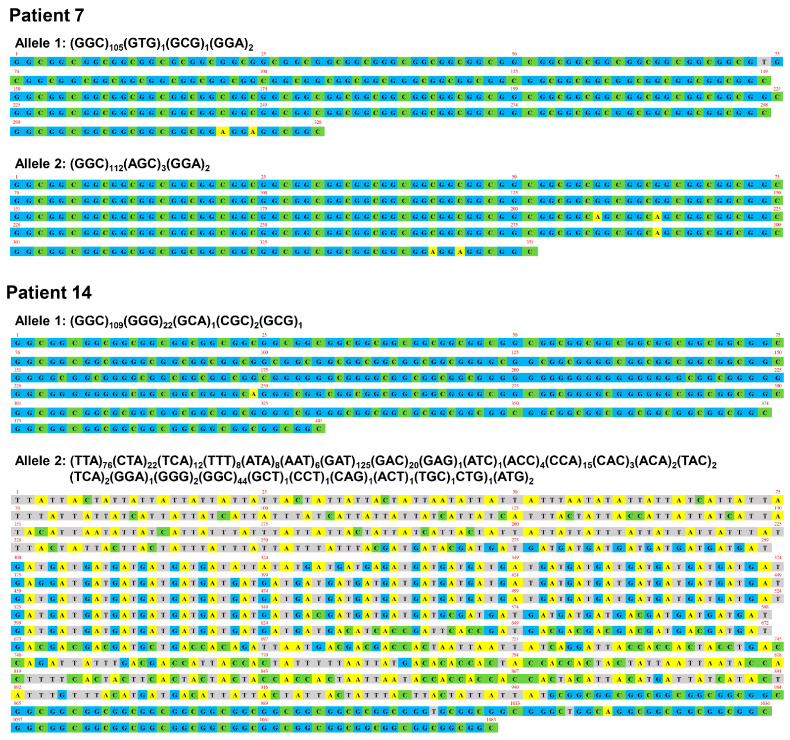
Expanded repeat sequences in *NOTCH2NLC*. Repeat length and sequence were determined by Nanopore long-read sequencer. In Pt 7, the expanded GGC repeats (n = 107) were followed by two GGA sequences in allele 1. In allele 2, the expanded GGC repeats (n = 112) were interrupted by three AGC motifs, followed by two GGA sequences. In Pt 14, GGG, GCA, CGC, and GCG sequences were inserted within the expanded GGC repeats (n = 109) in allele 1. In allele 2, a unique trinucleotide sequence containing expanded TTA (n = 76), GAT (n = 125), and GGC repeats (n = 44) was observed.

**Figure 2 brainsci-13-00955-f002:**
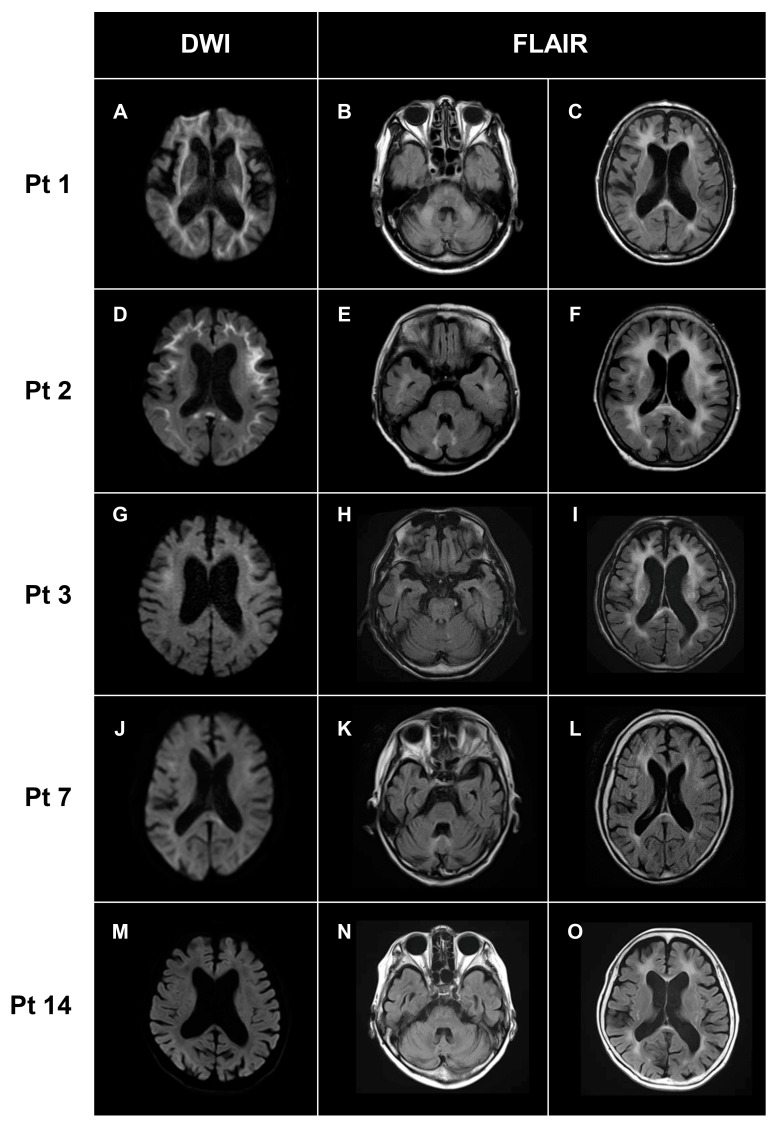
**Characteristic MRI findings.** Representative MRI images in Pt 1 (**A**–**C**), Pt 2 (**D**–**F**), Pt 3 (**G**–**I**), Pt 7 (**J**–**L**), and Pt 14 (**M**–**O**) were shown. High-intensity lesions of various degrees along with the U-fibers were observed on DWI (**A**,**D**,**G**,**J**,**M**). High-intensity lesions in the middle cerebellar peduncles (**B**) and paravermal lesions (**E**,**K**) appeared as cerebellar abnormalities on FLAIR. Bilateral white matter changes of various degrees were detected in all patients (**C**,**F**,**I**,**L**,**O**). DWI: diffusion-weighted image; FLAIR: fluid attenuated inversion recovery.

**Figure 3 brainsci-13-00955-f003:**
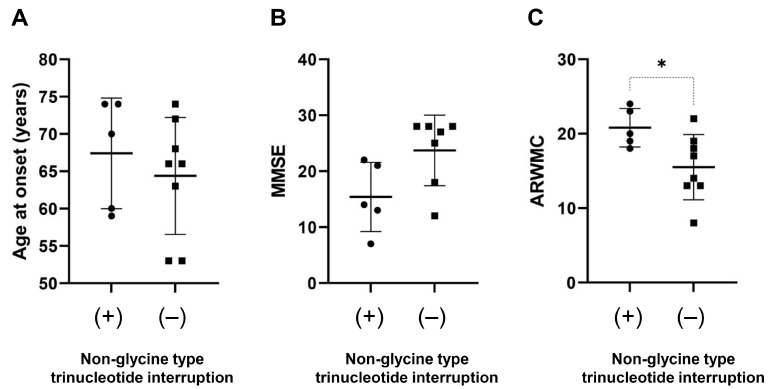
**Effect of non-glycine type trinucleotide interruption on phenotype**. Association of non-glycine type trinucleotide interruption with age at onset (**A**), MMSE (**B**), and ARWMC scores (**C**) was examined. Patients with non-glycine type trinucleotide interruption has a significantly larger ARWMC score than those without (**C**). *, *p* < 0.05.

**Table 1 brainsci-13-00955-t001:** Characteristics of *NOTCH2NLC* repeat expansions in patients with NIID.

	Pt 1	Pt 2	Pt 3	Pt 4	Pt 5	Pt 6	Pt 7	Pt 8	Pt 9	Pt 10	Pt 11	Pt 12	Pt 13	Pt 14	Pt 15
Sex	M	F	M	F	F	F	F	F	M	F	M	F	F	F	M
Age at onset	70	53	74	66	68	53	50	74	59	66	60	63	72	60	74
Age at examination	77	66	78	68	72	55	78	78	69	72	74	71	78	63	84
Disease duration	7	13	4	2	4	2	28	4	10	6	14	8	6	3	10
Family history	–	–	–	–	–	–	+	–	–	–	–	+	–	–	–
*NOTCH2NLC* repeat length															
Trinucleotide repeat size (Short allele)	19	23	20	30	20	20	N/A	21	15	20	20	22	20	N/A	15
GGC repeat size (Short allele)	18	18	18	29	18	18	N/A	19	12	18	18	20	18	N/A	12
Poly-glycine repeat size (Short allele)	19	20	20	30	20	20	N/A	21	15	20	20	22	20	N/A	15
Trinucleotide repeat size (Expanded allele)	198	104	158	102	101	107	109/117	101	109	113	175	141	94	135/361	97
GGC repeat size (Expanded allele)	193	104	156	101	99	106	105/112	90	101	110	168	140	92	109/44	95
Poly-glycine repeat size (Expanded allele)	196	104	156	102	101	107	108/114	92	103	113	169	141	94	131/47	97
GGC (%)	97.5	100.0	98.7	99.0	98.0	99.1	96.4/95.7	89.0	93.7	97.3	96.0	99.3	97.9	80.0/12.2	97.9
GGA (%)	1.0	0.0	0.0	1.0	2.0	0.0	1.8/1.7	2.0	1.8	1.8	0.6	0.7	2.1	0.0/0.3	2.1
GGG (%)	0.0	0.0	0.0	0.0	0.0	0.9	0.0	0.0	0.0	0.9	0.0	0.0	0.0	16.3/0.6	0.0
AGC (%)	1.5	0.0	1.3	0.0	0.0	0.0	0.0/2.6	6.0	0.0	0.0	1.1	0.0	0.0	0.0	0.0
GAC (%)	0.0	0.0	0.0	0.0	0.0	0.0	0.0	1.0	0.0	0.0	0.0	0.0	0.0	0.0/3.6	0.0
GAT (%)	0.0	0.0	0.0	0.0	0.0	0.0	0.0	0.0	0.0	0.0	0.0	0.0	0.0	0.0/34.6	0.0
TTA (%)	0.0	0.0	0.0	0.0	0.0	0.0	0.0	0.0	0.0	0.0	0.0	0.0	0.0	0.0/21.1	0.0
Others (%)	0.0	0.0	0.0	0.0	0.0	0.0	1.8/0.0	2.0	5.5	0.0	2.3	0.0	0.0	3.7/27.5	0.0

F, Female; M, Male; N/A, not available

**Table 2 brainsci-13-00955-t002:** Clinical features in patients with NIID.

	Pt 1	Pt 2	Pt 3	Pt 4	Pt 5	Pt 6	Pt 7	Pt 8	Pt 9	Pt 10	Pt 11	Pt 12	Pt 13	Pt 14	Pt 15	Frequency (%)
Clinical subtype	D	P	D	M	P	M	P	P	D	D	D	O	P	P	D	-
Cognitive decline	+	+	+	–	+	-	+	+	+	+	+	–	–	+	+	11/15 (73%)
MMSE	14	28	13	25	12	27	15	22	7	28	21	28	N/A	27	18	-
Psychiatric symptoms	+	+	+	–	–	–	–	–	+	+	–	–	–	+	+	7/15 (47%)
Consciousness disturbance	–	+	–	–	+	–	+	+	+	-	+	+	+	+	–	9/15 (60%)
Vomiting	–	+	–	–	–	–	–	–	+	+	–	–	+	–	–	4/15 (27%)
Aphasia	–	–	–	+	–	–	+	+	–	–	–	–	–	–	–	3/15 (20%)
Dysarthria	–	–	+	+	–	+	-	+	–	+	+	+	–	–	+	8/15 (53%)
Tremor	–	+	–	+	+	+	+	+	–	+	-	–	–	–	–	7/15 (47%)
Cerebellar ataxia	+	+	+	+	-	–	–	–	+	+	+	+	–	–	+	9/15 (60%)
Gait disturbance	+	+	+	+	-	-	–	–	+	+	+	–	–	–	+	8/15 (53%)
Sensory disturbance	–	–	–	+	–	–	–	–	–	+	-	–	–	–	–	2/15 (13%)
Hyporeflexia	+	+	–	+	–	–	+	+	N/A	+	+	+	–	+	+	10/14 (71%)
Urinary disturbance	+	+	+	–	–	–	+	–	–	–	+	+	–	–	+	7/15 (47%)
Muscle weakness	+	–	–	–	–	–	–	+	–	+	–	–	–	+	+	5/15 (33%)
Encephalitic episodes	–	+	–	–	+	–	+	+	+	–	+	–	+	+	–	8/15 (53%)
Dysphagia	+	–	+	–	+	–	–	–	+	–	–	–	–	–	–	4/15 (27%)
Myoclonus	–	–	–	–	–	–	–	–	+	–	–	–	–	–	–	1/15 (7%)
Constipation	+	+	–	–	–	–	–	–	–	–	+	–	–	–	–	3/15 (20%)

D, dementia-dominant; P: paroxysmal symptom-dominant; M, movement disorder-dominant; O: other subtype; MMSE: Mini-mental state examination; N/A, not available.

**Table 3 brainsci-13-00955-t003:** CSF biomarker profiles in patients with NIID.

Biomarkers	Cutoff Value	Pt 3	Pt 10	Pt 11
Aβ42	359.6 pg/mL	764.9	757.4	521.2
Aβ42/40 ratio	0.072	0.125	0.106	0.095
p-tau181	30.6 pg/mL	65.4	65.6	69.0
t-tau	105.3 pg/mL	179.5	104.0	116.1
NfL	2650 pg/mL	93251	4430	7253
AT(N) classification		A–T+(N)+	A–T+(N)+	A–T+(N)+

Aβ, amyloid-β; NfL, neurofilament light chain; cutoff values were determined in previous study (15).

**Table 4 brainsci-13-00955-t004:** MRI features of patients with NIID.

	Pt 1	Pt 2	Pt 3	Pt 4	Pt 5	Pt 6	Pt 7	Pt 8	Pt 9	Pt 10	Pt 11	Pt 12	Pt 13	Pt 14	Pt 15	Frequency (%)
White matter lesions	+	+	+	+	+	+	+	+	+	+	+	+	+	+	+	15/15 (100%)
DWI high-intensity lesions in U-fibers	+	+	+	+	+	+	+	+	+	+	+	+	–	+	+	14/15 (93%)
DWI high-intensity lesions in posterior lobe	+	–	+	–	+	–	+	–	+	–	+	+	–	–	-	6/15 (44%)
DWI high-intensity lesions in corpus callosum	–	+	-	+	+	+	-	+	+	+	+	+	–	–	+	10/15 (67%)
FLAIR high-intensity lesions in corpus callosum	+	+	+	+	+	-	-	+	+	+	+	–	–	–	+	14/15 (93%)
Cerebellar atrophy	+	–	–	+	+	–	+	+	+	-	+	+	+	+	+	11/15 (73%)
Middle cerebellar peduncle lesions	+	–	+	–	–	–	–	–	–	+	–	–	–	–	–	3/15 (3%)
Paravermal lesions	–	+	–	–	+	–	+	+	–	–	–	+	–	–	+	5/15 (33%)
ARWMC	23	22	24	14	19	13	13	18	19	13	20	8	18	16	17	

DWI, diffusion-weighted image; FLAIR, fluid attenuated inversion recovery; ARWMC, age-related white matter changes.

**Table 5 brainsci-13-00955-t005:** Association between expanded GGC repeat number in *NOTCH2NLC* and clinical/MRI features.

	Expanded GGC Repeat Number, Mean ± SD (Range)		*p* Value
	Present	Absent	
Clinical Symptoms			
Cognitive decline	115.7 ± 41.7 (44–194, n = 11)	110.0 ± 20.7 (93–140, n = 4)	0.87
Hyporeflexia	115.8 ± 42.3 (44–194, n = 10)	113.5 ± 28.8 (93–156, n = 4)	0.84
Consciousness disturbance	105.7 ± 34.3 (44–168, n = 9)	127.0 ± 39.4 (95–194, n = 6)	0.29
Cerebellar ataxia	129.9 ± 35.9 (9–194, n = 9)	90.7 ± 24.3 (44–112, n = 6)	0.046 *
Dysarthria	120.8 ± 29.7 (90–168, n = 8)	106.7 ± 44.5 (44–194, n = 7)	0.48
Encephalitic episodes	101.4 ± 33.9 (4–168, n = 8)	128.9 ± 36.3 (95–194, n = 7)	0.13
Gait disturbance	128.6 ± 38.1 (95–194, n = 8)	97.7 ± 28.9 (44–140, n = 7)	0.14
Tremor	103.1 ± 7.4 (90–112, n = 7)	123.9 ± 49.1 (44–194, n = 8)	0.63
Urinary dysfunction	138.4 ± 36.6 (95–194, n = 7)	93.0 ± 20.8 (44–110, n = 8)	0.01 *
Psychiatric symptoms	114.9 ± 47.8 (44–194, n = 7)	113.6 ± 26.9 (90–168, n = 8)	0.89
MRI findings			
Cerebellar atrophy	109.0 ± 39.8 (44–193, n = 11)	128.0 ± 24.1 (106–156, n = 4)	0.13
DWI high-intensity lesions in corpus callosum	112.7 ± 40.9 (44–193, n = 10)	116.8 ± 29.7 (90–156, n = 5)	0.93
DWI high-intensity lesions in posterior lobe	129.5 ± 40.5 (99–193, n = 6)	103.8 ± 31.8 (44–156, n = 9)	0.19
Paravermal lesion	100.0 ± 8.5 (90–112, n = 5)	121.1 ± 43.3 (44–193, n = 10)	0.24

*, statistically significant; DWI, diffusion-weighted image; FLAIR, fluid attenuated inversion recovery.

## Data Availability

The datasets presented in this study are available from the corresponding author on reasonable request.

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
