# Peer review of "Heterogenous Genetic, Clinical, and Imaging Features in Patients with Neuronal Intranuclear Inclusion Disease Carrying NOTCH2NLC Repeat Expansion"

_brainsci, 2023, doi:10.3390/brainsci13060955_

Round 1

Reviewer 1 Report

I have several comments and suggestions, which are as follows:

 Point 1: The Abstract has to be re-written to clarify the background of the study and the rational of the study.

Point 2: In the abstract first time appear GGC. What is GGC? It has to be clear to the readers.

Point 3: The introduction's last paragraphs must be re-written to discuss the conclusion and future study outcomes.

Point 4: What have the researchers done to minimize the recruitment bias in this study?

Point 5: This statement should be more explicit: "Fifteen patients clinically diagnosed with NIID between April 2018 and November 2022 were recruited from 6 medical institutes and genetically evaluated".

Point 6: What are the inclusion criteria for the patients?  It should be more straightforward for readers.

Point 7: Please clarify the meaning: An average of 96.1 gigabases was obtained for each library. What reflects 96.1 gigabases?

Point 8: What is the basis of the Mini-Mental State Examination?

Point 9: What do the researchers have to do to limit the limitations of this study in future? Add to the discussion.

Point 10: The conclusion should guide the researchers for future NIID patients with NOTCH2NLC studies.

Author Response

Comment 1: The Abstract has to be re-written to clarify the background of the study and the rational of the study.

Reply: Thank you very much for your time and effort to review our manuscript. We believe that your comments have helped us improve our manuscript. Point-by-point response to each of your comments is described below.

In accordance with the comments, we rewrote the Abstract to clarify the background and rational of the study in the revised manuscript (page 1, lines 23-25).

Comment 2: In the Abstract first time appear GGC. What is GGC? It has to be clear to the readers.

Reply: We added a more explanation on GGC in the Abstract of the revised manuscript (page 1, line 21).

Comment 3: The introduction's last paragraph must be re-written to discuss the conclusion and future study outcomes.

Reply: In accordance with the comment, we added the conclusion and future study outcomes in the last paragraph of the Introduction of the revised manuscript (page 2, lines 74-78).

Comment 4: What have the researchers done to minimize the recruitment bias in this study?

Reply: This is an important point. We recruited the participants using following inclusion criteria. Inclusion criteria included 1) age at onset ≧20 years, 2) presence of the white matter lesions shown in MRI or CT scan, and 3) the presence of neurological, cognitive, or psychiatric symptoms. To increase the specificity, subjects with the following conditions were excluded: 1) the white matter lesions secondary to demyelinating disease, infection, toxins, or neoplasm, 2) past history of cerebral ischemic infarctions or intracranial hemorrhage, and 3) presence of known mutations of white matter diseases such as CSF1R, NOTCH3, or HTRA1. However, there may be still recruitment bias as we recruited patients with white matter lesions on MRI. Patients with the muscle weakness or peripheral neuropathy not showing MRI abnormalities may be underrepresented. We described the recruitment method in the Method (page 3, lines 87-93) and the limitation in the Discussion (page 12, lines 324-326)

Comment 5: This statement should be more explicit: "Fifteen patients clinically diagnosis with NIID between April 2018 and November 2022 were recruited from 6 medical institutes and genetically evaluated.

Reply: To explicit the description, we re-wrote the sentences as follows. Fifteen patients who met the inclusion and exclusion criteria were recruited from 6 medical institutes between April 2018 and November 2022. All the patients underwent a neurological examination, routine blood testing, and neuroimaging evaluation (page 3, lines 83-86 ).

Comment 6: What are the inclusion criteria for the patients? It should be more straightforward for readers.

Reply: Inclusion criteria for this genetic analysis included 1) age at onset >20 years, 2) presence of the white matter lesions shown in MRI or CT scan, and 3) the presence of neurologic, cognitive, or psychiatric symptoms. To increase the specificity, we excluded the subjects with following conditions: 1) the white matter lesions secondary to demyelinating disease, infection, toxins, or neoplasm, 2) past history of stroke, and presence of known mutations of white matter disorders such as CSF1R, NOTCH3, and HTRA1. These sentences are now included in Methods section (page 3, line 83-86).

Comment 7: Please clarify the meaning: An average of 96.1 gigabases was obtained for each library. What reflects 96.1 gigabases?

Reply: To clarify the meaning, we re-wrote the following sentence: This genetic analysis yeilded an average of 96.1 gigabases of sequence data from each sample. (page 3, lines 119-120).

Comment 8: What is the basis of the Mini-Mental State Examination?

Reply: Mini-Mental State Examination is a screening cognitive battery with a maximum score of 30. We included this description in the revised manuscript (page 4, lines 136-137).

Comment 9: What do the researchers have to do to limit the limitations of this study in future? Add to the discussion.

Reply: Thank you for the suggestion. We added future efforts to minimize the limitations in the Discussion (page 12, lines 319-321 and 324-326).

Comment 10: The conclusion should be guide the researchers for future NIID patients with NOTCH2NLC studies.

Reply: Thank you for the suggestion. The clinical diagnosis of patients with NIID has been difficult because of a broad spectrum of clinical presentations. Our study showed the usefulness of genetic testing of repeat-primed and amplicon-length PCR followed by long-read sequence for the diagnosis of NIID. Ensuring a correct diagnosis by genetic analyses is important for better management of patients with NIID. We added these sentences in the Conclusion (page 12, lines 337-339)

Reviewer 2 Report

The present study, on the diagnosis of Neuronal Intranuclear Inclusion Disease Carrying 3 NOTCH2NLC Repeat Expansion, is a well designed study presenting pertinent data. Minor editing are required for this work:

- The aim of the study is not provided. The authors may need to clearly described in the abstract and introduction.

Author Response

Comment 1: The present study, on the diagnosis of Neuronal Intranuclear Inclusion Disease Carrying NTOTH2NLC Repeat Expansion, is a well designed study presenting data. Minor editing are required for this work.

Reply: Thank you for the positive comments. We very much appreciate your time and effort to review our manuscript. Point-by-point response to your comment is shown below.

Comment 2: The aim of the study is not provided. The authors may need to clearly describe in the Abstract and Introduction.

Reply: Thank you the suggestion. We have added the aim of the study in the Abstract and Introduction (page 1, lines 23-25; page 3, lines 73-75).

Reviewer 3 Report

Introduction:

The introduction is really clear and easy to read. I would suggest to state the hypothesis and the aims of this study. 

Methods materials:
Patients: please add which gender you choose. 

I would also recommend to include a table with the population's characteristics either in Materials and methods or Results section. 

Results:

Results are clearly presented and they are well described 

Discussion:

Please add/ develop the perspective of these findings, and what will they bring to patients. 

Thanks

Author Response

Comment 1: The introduction is really clear and easy to read. I would suggest to state the hypothesis and the aims of this study.

Reply: Thank you for the positive comment. We very much appreciate your time and effort to review our manuscript. Point-by-point response to each of your comments is described below. In accordance with the suggestion, we now included hypothesis and aims of this study in the Introduction (page 3, lines 73-75).

Comment 2: Please add which gender you choose.

Reply: In a recruitment process, the patients were recruited regardless of gender. As a result, we recruited 10 female and 5 male subjects. This sentence was added in the Methods (page 3, lines 86-87)

Comment 3: Results are clearly presented and they are well described.

Reply: Thank you for the positive comment.

Comment 4: Please add/develop the perspective of these findings, and what will they bring to patients.

Reply: Thank you for the suggestion. We added the perspective of our findings and what will they bring to patients in the Conclusion (page 12, lines 337-339).

Round 2

Reviewer 1 Report

Most of the suggestions have been incorporated by the authors in the revised manuscript. Therefore, no issue with considering it for publication.